# The Oncogenic Theory of Preeclampsia: Is Amniotic Mesenchymal Stem Cells-Derived PLAC1 Involved?

**DOI:** 10.3390/ijms24043612

**Published:** 2023-02-10

**Authors:** Massimo Conese, Ottavio Napolitano, Onofrio Laselva, Sante Di Gioia, Luigi Nappi, Luigia Trabace, Maria Matteo

**Affiliations:** 1Department of Clinical and Experimental Medicine, University of Foggia, Via Napoli 121, 71122 Foggia, Italy; 2Department of Medical and Surgical Sciences, University of Foggia, Via Napoli 121, 71122 Foggia, Italy

**Keywords:** preeclampsia, amniotic membrane, mesenchymal stem cells, PLAC1, trophoblast

## Abstract

The pathomechanisms of preeclampsia (PE), a complication of late pregnancy characterized by hypertension and proteinuria, and due to improper placentation, are not well known. Mesenchymal stem cells derived from the amniotic membrane (AMSCs) may play a role in PE pathogenesis as placental homeostasis regulators. PLACenta-specific protein 1 (PLAC1) is a transmembrane antigen involved in trophoblast proliferation that is found to be associated with cancer progression. We studied PLAC1 in human AMSCs obtained from control subjects (n = 4) and PE patients (n = 7), measuring the levels of mRNA expression (RT-PCR) and secreted protein (ELISA on conditioned medium). Lower levels of PLAC1 mRNA expression were observed in PE AMSCs as compared with Caco2 cells (positive controls), but not in non-PE AMSCs. PLAC1 antigen was detectable in conditioned medium obtained from PE AMSCs, whereas it was undetectable in that obtained from non-PE AMSCs. Our data suggest that abnormal shedding of PLAC1 from AMSC plasma membranes, likely by metalloproteinases, may contribute to trophoblast proliferation, supporting its role in the oncogenic theory of PE.

## 1. Introduction

Preeclampsia (PE), a dangerous complication of pregnancy, is due to impaired placentation. Multipotent mesenchymal stem cells (MSCs) have been investigated for their immunomodulatory roles and involvement in the pathophysiology of many diseases [1]; however, at present, it is barely known how MSCs can contribute to the pathogenesis of PE [2,3]. In this paper, we have considered human amniotic MSCs (hAMSCs) in the pathogenesis of PE as a source of PLACenta-specific protein 1 (PLAC1), an antigen involved in trophoblast proliferation.

### 1.1. The Oncogenic Theory of Preeclampsia: The Role of PLAC1 Antigen

Genes of the gametogenic and/or trophoblastic lineage are frequently ectopically activated and robustly expressed in human cancer. Based on phenotypical similarities between germ cells, pregnancy trophoblasts, and cancer cells, as early as 100 years ago, John Beard proposed a “trophoblastic theory of cancer” [4,5]. PLAC1 is among the genes overexpressed in human cancer and is selectively expressed in the placental syncytiotrophoblast in adult normal tissues, playing an essential role in normal placental and embryonic development. Accumulating evidence suggests that enhanced PLAC1 expression is closely associated with the progression of human cancer; accordingly, the upregulated expression of PLAC1 in human hepatocellular cancer (HCC) was found significantly correlated with metastasis of HCC [6].

PLAC1 is a small (212 amino acids) cell surface protein; the encoding gene is present on the long arm of chromosome X. The PLAC1 protein contains three conserved functional domains: a signal peptide (aa 1–23), a transmembrane domain (aa 20–50), and the zona pellucida ZP3 domain (aa 58–118), which bears significant homology to the N-terminal subdomain of sperm receptor [7], suggesting that PLAC1 may participate in forming or stabilizing the fetus–mother interface, perhaps as a receptor [8]. PLAC1 mRNA expression is restricted to cells of trophoblastic origin, implicating a crucial factor in normal placental establishment and maintenance. PLAC1 polypeptide localizes to the differentiated syncytiotrophoblast throughout gestation (8–41 weeks), as well as to a small population of villous cytotrophoblasts [9,10]. Importantly, the syncytial staining appeared more intense along the microvillous brush border membranes, a finding further corroborated by deconvolution microscopy, which showed that PLAC1 localizes to cytosolic sites in close proximity to the apical plasma membrane [10,11].

We suggest that the aberrant and enhanced activity of PLAC1 protein could be associated with enhanced trophoblast proliferation and, as a consequence, with adverse pregnancy outcome. PE is a disorder characterized by hypertension and proteinuria after 20 weeks of gestation; it is sometimes accompanied by maternal multi-organ dysfunction, affects 3% to 5% of all pregnancies, and is one of the major causes for maternal and infant mortality and morbidity in the perinatal period [12]. The pathogenesis of preeclampsia is not completely known. However, the placenta plays a crucial role in the development of this complication [12]. The proliferation of non-invading trophoblasts, an excessive number of trophoblasts with a high proliferative index but low phenotypic maturity and increased trophoblast apoptosis, was noticed in preeclamptic placentas [13,14].

### 1.2. Amniotic Mesenchymal Stem Cells

On this basis, we aimed to evaluate the expression of PLAC1 antigen in placentas from preeclamptic patients. We chose hAMSCs to study this issue because they are derived from mesoderms, which, in turn, generates the cytotrophoblast layer of placenta. It is well known that the cytotrophoblasts completely replace the syncytiotrophoblast after 20 weeks of pregnancy, and that PE only develops after the 20th week of gestation. Among the different types of MSCs present in the placenta [15], hAMSCs can be harvested from the mesenchymal layer of the amniotic membrane by a simple enzymatic digestion procedure, and a high yield of cells can be achieved [16]. hAMSCs are easily cultured on plastic and can be expanded for many passages without losing their morphology [17]. These characteristics make it possible to use high amounts of these cells for studying their functional features. The relationship among hAMSCs, trophoblast invasion, and preeclampsia, although rationally intriguing, is unknown.

## 2. Results

### 2.1. Clinical Characteristics of Patients

The clinical data of enrolled subjects are reported in Table 1. There were no significant differences in maternal age, gestational age at delivery, and neonatal birth weight between control and preeclamptic patients. There was a significant difference in blood pressure and proteinuria that resulted significantly higher in preeclamptic patients.

### 2.2. hAMSC Yield, Morphology, and Phenotype

The yield of hAMSCs from preeclamptic amniotic membranes (each of 5 cm^2^) was comparable with that of non preeclamptic ones (preeclamptic: 6.51 ± 3.98 × 10^6^; non-preeclamptic: 7.71 ± 5.33 × 10^6^ [mean ± SD]). As indicated by the International Society for Cellular Therapy (ISCT), which proposed minimal criteria for identification of human MSCs [18], hAMSCs presented a fibroblast-like morphology in culture and were confirmed to be CD45 (−), CD11b (−), CD90 (+), CD73 (+), and CD105 (+) via flow cytometry [19].

### 2.3. PLAC1 mRNA Expression Tends to Be Lower in Preeclamptic Patients

Since PLAC1 mRNA expression level has not been previously investigated in hAMSCs, we employed, in preliminary experiments, two sets of specific primers for RT-PCR studies [20,21]. Moreover, since it has been demonstrated that the CaCo2 cell line expresses PLAC1 [22,23], we used CaCo2 cells as a positive control. CaCo2 cells showed higher levels than those detectable in non-preeclamptic and preeclamptic hAMSCs with both set of primers (Figure 1a,b). Since the Δct values for primer set #2 were lower, we used the primer set #1 for the subsequent evaluation of PLAC1 mRNA. By analyzing a subset of hAMSCs from non preeclamptic and control placentae, lower levels of PLAC1 mRNA expression were observed in preeclamptic hAMSCs as compared with Caco2 cells but not with controls (Figure 1c).

### 2.4. Secreted PLAC1 Antigen Is Detectable in Preeclamptic hAMSCs but Not in Controls

To evaluate the capacity of hAMSCs to secrete the PLAC1 antigen, conditioned medium was assayed by an ELISA kit for the same hAMSCs used in the RT-PCR analysis. As Figure 2 shows, conditioned medium obtained from preeclamptic hAMSCs presented detectable levels of PLAC1, whereas the PLAC1 antigen was undetectable in conditioned medium from control hAMSCs (*p* = 0.0130).

### 2.5. Analys of PLAC1 Cleavage Sites Identifies Many Proteases That Could Shed PLAC1 Antigen

The presence of PLAC1 antigen, a membrane protein [10], in the conditioned medium of preeclamptic hAMSCs strongly suggests that PLAC1 may be shed via endogenous proteases. In silico analysis of cleavage sites in the primary sequence of PLAC1 allowed the identification of those proteases acting on its extracellular domain: HIV-1 retropspesin, cathepsin K, matrix metalloprotease (MMP)-2, MMP-9, MMP-3, chymotrypsin A (cattle-type), elastase-2, cathepsin G, glutamyl peptidase-I, thylakoidal processing peptidase. Considering that the PLAC1 transmembrane domain ends at position 50, and for those results with biological plausibility, one should consider the enzymes listed in Table 2 that could be involved in PLAC1 shedding.

## 3. Discussion

This study evaluated, for the first time, the expression of the PLAC1 protein in hAMSCs from placentas derived from preeclamptic patients. The results demonstrated a significantly higher secretion of PLAC1 antigen in the conditioned medium of the hAMSCs of preeclamptic women compared to controls, although the difference between mRNA expression determined by RT-PCR analysis did not differ significantly between preeclamptic patients and controls.

Placenta-derived MSCs (PMSCs) have been considered an ethical and valid alternative to bone marrow-derived MSCs in research related to this cell type, due to their abundance, ease of availability, low immunogenicity and effective immunomodulatory ability [24,25], long-term growth ability [26], slow aging rate [27], and potent expansion ability [28]. PMSCs are closely involved in placentation through differentiation into endothelial cells to initiate early placental vasculogenesis [29]. Moreover, PMSCs can improve the placental angiogenesis and the uterine spiral artery remolding [30,31], as well as to modulate the uteroplacental immune status [32], through both cell–cell contact and paracrine signaling. Although the role of PMSCs in PE pathogenesis has not been completely discovered yet, recent work highlights that PMSCs may be involved in the development and severity of PE due to their dysfunctional and senescent status with detrimental paracrine action towards other cell types (reviewed in [3]). In this communication, we propose that hAMSCs might be involved in another mechanism related to PLAC1.

hAMSCs derive from the mesoderm, which in turn generates the cytotrophoblast layer of placenta. The presence of PLAC1 protein in hAMSCs is in accordance with a previous study reporting that at 8 weeks of gestation, PLAC1 localizes primarily to the differentiated syncytiotrophoblast, whereas the underlying cytotrophoblast contains little or no protein [10]. At 23 weeks, however, PLAC1 was observed in specific immunostaining of the cytotrophoblast [10], suggesting a possible role of this protein for placental development, not only in the early stage of the implantation process but in all phases of pregnancy. Accordingly, the expression of the PLAC1 antigen was found to be significantly different in the case of complicated pregnancies compared to controls. A decrease in pregnancy-associated plasma protein A (PAPPA), a syncytiotrophoblast-derived metalloproteinase, and, in accordance with our results, an increase in PLAC1 expression was demonstrated in pregnancy complicated by preeclampsia as compared with non-PE controls [21]. Moreover, increased plasma levels of PLAC1 mRNA were observed in the preeclamptic patients and were explained by the damage to placental villi and the apoptosis of trophoblast cells due to the oxidative stress associated with preeclampsia. On the contrary, PLAC1 mRNA levels, dramatically decline in maternal plasma, at the first day after delivery, and could not be detected one month later [33], confirming its placental origin.

The presence of PLAC1 in the conditioned medium of hAMSCs obtained from women with preeclampsia and its complete absence in the controls, strongly indicates that PLAC1 is a membrane protein and that it can be released from the cell by endogenous cellular proteases as a consequence of placental damage. Since PLAC1 expression is closely associated with proliferation activity and progression in human cancer, we suggest that a dysregulation of cytotrophoblastic cells and an aberrant shedding of PLAC1 antigen, as above described, could in turn be responsible for the enhanced trophoblast proliferation and the excessive collagen deposition associated with preeclampsia. In accordance with our hypothesis, increased trophoblast proliferation in preeclamptic placentas has been reported by some investigators [34]. For example, although the overall volume of trophoblasts and the exchange surface area are not significantly altered in preeclampsia [35], a higher rate of proliferation in villous trophoblasts was observed by Redline et al. [13]. This study examined the implantation site of normal and preeclamptic placentas from various gestational ages ranging from 25 to 40 weeks using various cell proliferation and phenotypic markers. An excessive number of trophoblasts with a high proliferative index, but low phenotypic maturity, were detected in the preeclamptic placenta [13]. We hypothesized that, if PLAC1 antigen is involved in the aberrant proliferation of tumor cells so as to hypothesize a “trophoblastic theory of cancer”, an aberrant secretion of PLAC1 might be similarly responsible for a behavior analogous to that of cancer in the trophoblast of preeclamptic placentas, and, as a consequence, an “oncogenic theory of preeclampsia” should be considered.

Table 2 reports enzymes that could be involved in PLAC1 shedding from hAMSCs. Among the enzymes listed are matrix metalloprotease MMP-2 and MMP-9. MMP-2 (gelatinase A) and MMP-9 (gelatinase B) play an important role in endometrial tissue remodeling during the menstrual cycle and pregnancy [36], since trophoblast invasion into the decidual stroma may require degradation of ECM proteins. In fact, MMP-2 and MMP-9 are abundantly expressed in invading extravillous trophoblast cells, and the expression of these two gelatinases is highly related to trophoblast cell invasiveness [37].

Additionally, factors that promote trophoblast invasion appear to have regulatory effects on MMP-2 and/or MMP-9 activity. Genetic polymorphisms in MMP-2 and MMP-9 transcription have been described in preeclampsia [38]. We suggest that the aberrant functioning of MMPs could cause an abnormal PLAC1 antigen shedding, which could in turn be responsible for the enhanced trophoblast proliferation and the excessive collagen deposition previously found to be associated with MMP defects and preeclampsia.

Results from PCR analysis revealed that the difference in mRNA expression level between placenta from preeclamptic patient and controls did not reach statistical significance, confirming the hypothesis that the abnormal presence of PLAC1 protein in a subset of preeclamptic patients, could be a consequence of irregular shedding instead of increased production. Notably, the mechanism of abnormal shedding of other brush border-localized proteins has been previously reported by other authors in patients affected by preeclampsia, with an excess of protein in the serum of preeclamptic patients compared to controls associated with a decrease in their mRNA expression [39].

## 4. Materials and Methods

### 4.1. Subjects

Placental tissues were collected after caesarean section from 4 women with normal pregnancy and 7 women with preeclampsia. The study was approved from the Institutional Ethical Committee of University Hospital of Foggia (n. 138/CE/2020).

Preeclampsia was defined as hypertension that developed after GW20 (systolic or diastolic blood pressure ≥140 or ≥90 mmHg, respectively, measured at two different time points, 4 h to 1 week apart) coupled with proteinuria (≥300 mg in a 24 h urine collection, or two random urine specimens obtained 4 h to 1 week apart containing ≥1+ by dipstick or one dipstick of ≥2+ protein), according to the criteria recommended by the American College of Obstetricians and Gynecologists [40].

### 4.2. Cell Cultures

hAMSCs were isolated from a piece of amniotic membrane of 5 cm^2^, and grown in advanced DMEM supplemented with 10% FBS, 55 μM β-mercaptoethanol, 1% L-glutamine, 1% penicillin/streptomycin, and 10 ng/mL epidermal growth factor (EGF) (Sigma-Aldrich, Milan, Italy), as previously described [19,41]. hAMSCs were subcultured until passage 2 and analyzed for their phenotype [19]. Passage 2 hAMSCs were seeded onto 60 mm Petri dishes at a cell density of 5 × 10^5^/dish. Once cells reached 80% of confluence, medium was exchanged for fresh solution and collected after 24 h. Cells were used for RNA extraction.

Colon cancer CaCo2 cells were grown using high-glucose and L-glutamine Dulbecco’s modified Eagle medium (DMEM) medium supplemented with 10% fetal bovine serum (FBS), 1% penicillin and streptomycin [42]. They were subjected to the same procedure as hAMSCs.

The usage of hAMSCs was conducted according to the guidelines of the Declaration of Helsinki and approved by the Ethics Committee of the University Hospital of Foggia (n. 138/CE/2020 issued on 30 November 2020). Informed consent was obtained from all women included in the study.

### 4.3. Ribonucleic Acid Extraction and Quantification Reverse Transcription Polymerase Chain Reaction

RNA was extracted according to the manufacturer’s protocol (Direct-zol RNA MiniPrep; Zymo Research, Irvine, CA, USA). Total RNA was converted to cDNA with iSCRIPT™ cDNA synthesis kit (Bio-Rad, Hercules, CA, USA) and quantitative real-time PCR was performed using SsoAdvanced Universal SYBR Green Supermix (Bio-Rad, Hercules, CA, USA) [43]. Primers for PLAC1 and GAPDH are listed in Table 3. The quantification of the relative gene levels was calculated using the 2^−ΔΔCt^ method.

### 4.4. PLAC1 Secretion

hAMSC conditioned medium was centrifuged at 200× *g* for 10 min and supernatants were assayed by the ELISA Kit for human PLAC1 (Cat # MBS280640, MyBiosource, San Diego, CA, USA). The minimum detectable dose of human PLAC1 was determined to be 0.078 ng/mL. As background for the assay, medium with culture medium was tested and subtracted from samples. For each subject, 3 dishes were tested with 2 technical replicates, having for 4 control hAMSCs (n = 24 individual values) and for 7 pre-eclamptic hAMSCs (n = 42 values).

### 4.5. In Silico Analysis

To predict PLAC1 cleavage sites of multiple proteases, PLAC1 primary sequence (MKVFKFIGLMILLTSAFSAGSGQSPMTVLCSIDWFMVTVHPFMLNNDVCVHFHELHLGLGCPPNHVQPHAYQFTYRVTECGIRAKAVSQDMVIYSTEIHYSSKGTPSKFVIPVSCAAPQKSPWLTKPCSMRVASKSRATAQKDEKCYEVFSLSQSSQRPNCDCPPCVFSEEEHTQVPCHQAGAQEAQPLQPSHFLDISEDWSLHTDDMIGSM) was analyzed through PROSPER (Protease specificity prediction server; https://prosper.erc.monash.edu.au/home.html, accessed on 4 February 2023), allowing to obtain site cleavage predictivity by major protease families: aspartic proteases, cysteine proteases, metalloproteases, serine proteases. To avoid false positives, a higher threshold value of 0.8 for the “cleavage probability score” was used, as suggested [45].

### 4.6. Statistical Analysis

Statistical analyses were carried out by Prism v. 5.0 (GraphPad Software Inc., La Jolla, CA, USA). Data were expressed as mean ± SD or median and interquartile range. For clinical parameters statistical significance was evaluated by a two-tailed unpaired Student’s *t*-test. PLAC1 secretion data were evaluated by Mann–Whitney test. Multiple comparisons were based on one-way analysis of variance (ANOVA) with Tukey’s post hoc test. Significant differences were obtained when *p* < 0.05.

## 5. Conclusions

We can conclude that an abnormal excretion of PLAC1 antigen from mesoderm-derived cells (hAMSCs or cytotrophoblastic cells), was demonstrated in placenta from preeclamptic patients.

This significantly higher concentration of PLAC1 protein could derive from an abnormal shedding of PLAC1 antigen, probably related to an aberrant function of proteolytic enzymes, MMP-2 and/or MMP-9, associated with preeclampsia as previously reported in literature.

Further investigations are in progress in order to confirm the role of MMPs in the abnormal PLAC1 protein shedding and to evaluate the histological changes in the placenta in this subset of preeclamptic patients.

## Figures and Tables

**Figure 1 ijms-24-03612-f001:**
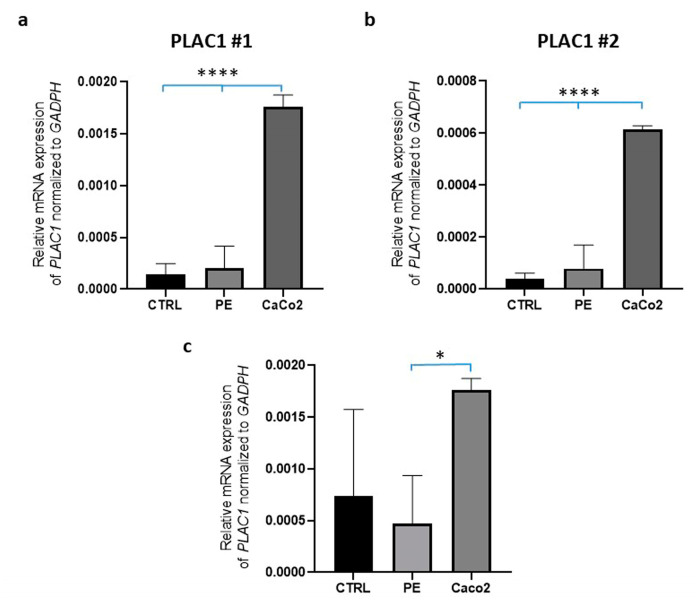
PLAC1 mRNA expression in preeclamptic and control hAMSCs. The expression levels were evaluated by RT-PCR analysis and normalized to the housekeeping gene GAPDH. (**a**,**b**) PLAC1 mRNA expression was assayed by primer set PLAC1 #1 and PLAC1 #2 in non-preeclamptic hAMSCs (CTRL; n = 2), preeclamptic hAMSCs (PE; n = 2), and CaCo2 cells, respectively. (**c**) PLAC mRNA expression was assayed by primer set PLAC #1 in non-preeclamptic (CTRL; n = 4), preeclamptic (PE; n = 7) hAMSCs, and in CaCo2 cells. Data are expressed as mean ± SD and were obtained from n = 3 biological replicates, each tested in duplicate. Statistical differences were determined using ANOVA with Tukey’s post hoc test. * *p* < 0.05; **** *p* < 0.0001.

**Figure 2 ijms-24-03612-f002:**
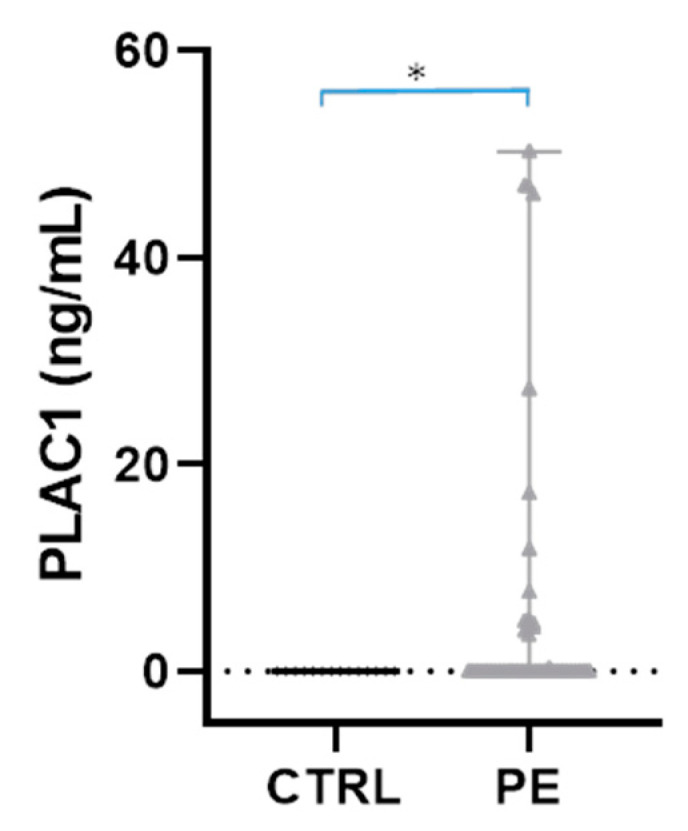
PLAC 1 antigen expression in control and preeclamptic hAMSCs. Conditioned medium from control (CTRL; n = 4) and preeclamptic (PE; n = 7) hAMSCs was assayed for PLAC1 by ELISA. Data are shown as individual well values (which for CTRL are stacked all on the 0 line), median and range. * *p* < 0.05 by Mann–Whitney test.

**Table 1 ijms-24-03612-t001:** Clinical characteristics of preeclamptic and control subjects (mean [SD]; median [interquartile range]).

	Preeclamptic (n = 7)	Controls (n = 4)	*p*
Maternal age (years)	32.0 (5.2); 32 (26.0–36.0)	34.0 (5.2); 34 (29.2–38.7)	0.55
Gestational age at delivery (weeks)	37.3 (2.7); 37.0 (36.0–40.0)	38.2 (0.5); 38.0 (38.0–38.7)	0.50
Systolic BP (mmHg)	146.0 (6.1); 145.0 (140.0–150.0)	122.5 (2.9); 122.5 (120.0–125.0)	**<0.0001**
Diastolic BP (mmHg)	95.0 (4.1); 95.0 (90.0–100.0)	80.0 (4.1); 80.0 (76.2–85)	**0.0002**
Birth weight (grams)	2759.0 (620.3); 2780.0 (2480.0–3450.0)	3135.0 (188.4); 3115.0 (2973.0–3318.0)	0.2762
Proteinuria (mg/24 h)	542.3 (211.9); 480.0 (315.0–735.0)	0 (0); 0 (0–0)	**0.0007**

Statistical differences were determined using the two-tailed unpaired Student’s *t* test. Significant values, i.e., *p* < 0.05, have been indicated in bold.

**Table 2 ijms-24-03612-t002:** PLAC1 cleavage sites of multiple enzymes.

Protease	Cleavage Site	Sequence	Aminoterminal Fragment	Carboxyterminal Fragment
Cystein protease				
Cathepsin K	53	VHFH  ELHL	6.27 kDa	18.35 kDa
	196	HFLD  ISED	22.69 kDa	1.93 kDa
Metalloprotease				
MMP-2	188	EAQP  LQPS	21.75 kDa	2.87 kDa
MMP-3	85	IRAK  AVSQ	10.23 kDa	14.38 kDa
	108	TPSK  FVIP	12.86 kDa	11.76 kDa
	118	CAAP  QKSP	13.84 kDa	10.77 kDa
	207	HTDD  MIGS	23.99 kDa	0.63 kDa
MMP-9	54	HFHE  LHLG	6.40 kDa	18.22 kDa
	64	CPPN  HVQP	7.63 kDa	16.98 kDa
	70	QPHA  YQFT	8.30 kDa	16.31 kDa
	72	HAYQ  FTYR	8.59 kDa	16.02 kDa
	76	FTYR  VTEC	9.16 kDa	15.46 kDa
	90	VSQD  MVIY	10.73 kDa	13.88 kDa
	108	TPSK  FVIP	12.86 kDa	11.76 kDa
	129	KPCS  MRVA	15.10 kDa	9.52 kDa
	179	VPCH  QAGA	20.75 kDa	3.86 kDa
	188	EAQP  LQPS	21.75 kDa	2.87 kDa
	207	HTDD  MIGS	23.99 kDa	0.63 kDa
Serin protease				
Elastase 2	82	ECGI  RAKA	9.88 kDa	14.74 kDa
	92	QDMV  IYST	10.97 kDa	13.65 kDa
	98	STEI  HYSS	11.67 kDa	12.94 kDa
	174	EEHT  QVPC	20.19 kDa	4.43 kDa
Cathepsin G	52	CVHF  HELH	6.13 kDa	18.48 kDa
	94	MVIY  STEI	11.24 kDa	13.38 kDa
	100	EIHY  SSKG	11.97 kDa	12.64 kDa
	130	PCSM  RVAS	15.23 kDa	9.39 kDa
	150	YEVF  SLSQ	17.53 kDa	7.09 kDa
	168	PCVF  SEEE	19.48 kDa	5.14 kDa
	194	PSHF  LDIS	22.46 kDa	2.16 kDa
Glutamyl peptidase-I	148	KCYE  VFSL	17.28 kDa	7.33 kDa

Red rectangles indicate the cleavage site.

**Table 3 ijms-24-03612-t003:** RT-PCR primers used in this study.

Primers	Sequence	Reference
PLAC1 #1 forward	5′-ATTGGCTGCAGGGATGAAAG-3′	[21]
PLAC1 #1 reverse	5′-TGCACTGTGACCATGAACCA-3′	
PLAC1 #2 forward	5′-CAGTGAGCACAAAGCCACATTTC-3′	[20]
PLAC1 #2 reverse	5′-CCACATTCAGTAACACGGTAGGTG-3′	
GAPDH forward	5′-CAAGAGCACAAG AGGAAGAGAG-3′	[44]
GAPDH reverse	5′-CTACATGGCAACTGTGAGGAG-3′	

## Data Availability

Not applicable.

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
