# Peer review of "The Oncogenic Theory of Preeclampsia: Is Amniotic Mesenchymal Stem Cells-Derived PLAC1 Involved?"

_ijms, 2023, doi:10.3390/ijms24043612_

Round 1

Reviewer 1 Report

- Population samples are extremely low

- Statistics done by group 7 compared to four?!

-Novel topic like this requires also more detailed description

Author Response

 Population samples are extremely low

- Statistics done by group 7 compared to four?!

-Novel topic like this requires also more detailed description.

We agree with the Reviewer’s concern about the low numbers of patients and controls presented in our manuscript, which should be considered a proof-of-concept study of a novelty in the field of preeclampsia, as also appreciated by the Reviewer. We are processing other samples from preecamptic patients and control subjects to confirm these data.

To comply with the Reviewer’s observation, we have added in the introduction a novel paragraph describing PLAC1 protein and its epression levels, as well as its putative role(s) in placentation.

Reviewer 2 Report

This study firstly investigated the role of amniotic mesenchy-2 mal stem cells-derived PLAC 1 in oncogenic theory of preeclampsia. The authors found hAMSC yield, morphology and phenotype. They also found the PLAC 1 mRNA expression and PLAC 1 antigen secretion. Furthermore, they predicted the PLAC 1 cleavage sites. Considering the clinical significance, this study can be considered to be published. However, several concerns still need to be revised before publication.

Minor revision:

1.     In the discussion part, the authors should fully summarize the current evidence about the oncogenic theory of preeclampsia and its effect on preeclampsia research.

2.     In the figure legends, the authors should illustrate how many independent experiments were done.

3.     There are some grammatical errors that need to be revised.

4.     The authors should summarize the abbreviation to increase the readability.

5.     In the statistical analyses part, the author should clearly indicate two-tail or one tail P value.

Author Response

This study firstly investigated the role of amniotic mesenchy-2 mal stem cells-derived PLAC 1 in oncogenic theory of preeclampsia. The authors found hAMSC yield, morphology and phenotype. They also found the PLAC 1 mRNA expression and PLAC 1 antigen secretion. Furthermore, they predicted the PLAC 1 cleavage sites. Considering the clinical significance, this study can be considered to be published. However, several concerns still need to be revised before publication.

Minor revision:

  1. In the discussion part, the authors should fully summarize the current evidence about the oncogenic theory of preeclampsia and its effect on preeclampsia research.

      We agree with the Reviewer's observation but there is no evidence in the literature on the “oncogenic theory of preeclampsia”.  It is a new pathogenetic hypothesis that the authors have proposed in this study, on the model of what has already been theorized by John Beard as much as 100 years ago as “trophoblastic theory of cancer”.

      The only evidence currently in the literature is that relating to the aberrant proliferation of the trophoblast of the preeclamptic placentas as reported on page 2, section 1.1 and on page 7  in  the  Discussion  of the manuscript.  The authors hypothesized in this manuscript that the trophoblast of preeclamptic patients may have a behavior analogous to that of cancer with an uncontrolled proliferation due to an aberrant shedding of the PLAC1 antigen.

       Anyway we agree with reviewer that this point should be highlighted, and we added in the Discussion the following sentence:

      “We hypothesized that, if PLAC 1 antigen is involved in the aberrant proliferation of tumor cells so as to hypothesize a “trophoblastic theory of cancer”, similarly an aberrant secretion of PLAC 1 could be responsible for a behavior analogous to that of cancer in the trophoblast of preeclamptic placentas, and, as a consequence, an “oncogenic theory of preeclampsia” should be considered”.

  1. In the figure legends, the authors should illustrate how many independent experiments were done.

      We have highlighted in the legend to Figure 1, that three independent biological replicates, each studied as two technical replicates, were done. In Figure 2, the data were obtained from the same samples employed in Figure 1, i.e. three dishes were tested for each subject with two technical replicates.

  1. There are some grammatical errors that need to be revised.

We have thoroughly revised the text to avoid grammatical errors.

  1. The authors should summarize the abbreviation to increase the readability.

We have added a list of the most frequent abbreviations just before the References.

  1. In the statistical analyses part, the author should clearly indicate two-tail or one tail P value.

We have clearly indicated in Table 1 that a two-tailed unpaired Student’s t test was used.

Reviewer 3 Report

In the manuscript " The Oncogenic Theory of Preeclampsia: Is Amniotic Mesenchy-mal Stem Cells-Derived PLAC 1 involved?" authors hypothesize the involvement of PLAC1 in trophoblast proliferation and support the PE-related oncogenic theory. 

The topic is interesting, but it is necessary to make some changes.

-Improve the resolution of figure 1.

-Are the same samples analyzed in fig 1A, 1B, and 1C?

- I suggest changing the titles of the results section using as titles the result obtained from the experiment.

- Supplementary Table 1 could be inserted into the text.

- it would also be required to see the expression protein PLAC1 protein by western blot assays in hAMSCs compared with control.

- In discussion, lane 142, you have to replace the word: expressioa... with expression....; lane 145: their their is twice repeated; lane 204: an an enhanced....an is repeated.

-Always write the same way PLAC1 without a space (PLAC 1).

_ Finally, in addition to PLAC1 expression, it would be useful to include markers of neoplasia, inflammation, and cell differentiation.

Author Response

In the manuscript " The Oncogenic Theory of Preeclampsia: Is Amniotic Mesenchy-mal Stem Cells-Derived PLAC 1 involved?" authors hypothesize the involvement of PLAC1 in trophoblast proliferation and support the PE-related oncogenic theory. 

The topic is interesting, but it is necessary to make some changes.

-Improve the resolution of figure 1.

We have now improved the resolution of Figure 1.

-Are the same samples analyzed in fig 1A, 1B, and 1C?

As we have explained in the Figure Legend, Figure 1a and b represent data obtained from a subset of patients (n=2) and controls (n=2), whereas Figure 1c is obtained from the analysis of all patients (n=7) and all controls (n=4).

- I suggest changing the titles of the results section using as titles the result obtained from the experiment

In agreement with the Reviewer’s proposal, we changed some of the titles of the results section to convey the results obtained from the experiment.

- Supplementary Table 1 could be inserted into the text.

We inserted the Supplementary Table 1 into the text, now it is Table 2.

- it would also be required to see the expression protein PLAC1 protein by western blot assays in hAMSCs compared with control.

It has been previously shown that another border brush protein of syncithiotrophoblast, i.e. Placental protein 13 (galectin-13), exactly as PLAC1 in this paper, is lower in mRNA expression and higher in serum of preeclamptic patients as compared with controls (Than et al., Virchows Arch (2008) 453:387–400; ref. 39 of the revised manuscript). Nevertheless, Western blotting did not detect any change in Placental protein 13 expression, indicating that an abnormal sheddase activity is responsible for this dicothomy in expression and secretions levels. For this reason, we acknowledge the point raised by the Reviewer, but we think that, in line with the previous cited study, that protein expression study in hAMSCs would be not decisive in the comprehension of the process of PLAC1 protein data as secreted antigen.

- In discussion, lane 142, you have to replace the word: expressioa... with expression....; lane 145: their their is twice repeated; lane 204: an an enhanced....an is repeated.

We have corrected these typos.

-Always write the same way PLAC1 without a space (PLAC 1).

We have always written PLAC1 without a space.

_ Finally, in addition to PLAC1 expression, it would be useful to include markers of neoplasia, inflammation, and cell differentiation.

We agree with the Reviewer that the role of PLAC1 in the oncogenic theory of preeclampsia, as proposed in this short communication, would have been more valuable if markers of placental neoplasia, inflammation and cell differentiation were presented, The fact is that we did not store the placental tissues used in this study to isolate hAMSCs, and this is why we are processing another cohort of patients and control subjects to verify hAMSCs results and to study those markers in placentas.

Round 2

Reviewer 1 Report

Thank you and appreciate your reply. The article should be marked as a pilot study thereof. In this form I have no problem with publishing of the article.

Kind regards,

Reviewer1

Reviewer 3 Report

The Authors have clearly replied point by point to my suggestions, and although it should still be implemented by further and future experiments, in its present form the manuscript can be published.